# Effect of Pre-Hospital Intubation in Patients with Severe Traumatic Brain Injury on Outcome: A Prospective Cohort Study

**DOI:** 10.3390/jcm8040470

**Published:** 2019-04-06

**Authors:** Caroline Choffat, Cecile Delhumeau, Nicolas Fournier, Patrick Schoettker

**Affiliations:** Service of Anesthesiology, University Hospital of Lausanne, 1011 Lausanne, Switzerland; cecile.delhumeau@hcuge.ch (C.D.); nicolas.fournier@gmail.com (N.F.); patrick.schoettker@chuv.ch (P.S.)

**Keywords:** neurotrauma, severe traumatic brain injury, pre-hospital intubation, secondary injury

## Abstract

Secondary injuries are associated with bad outcomes in the case of severe traumatic brain injury (sTBI). Patients with a Glasgow Coma Scale (GCS) < 9 should undergo pre-hospital intubation (PHI). There is controversy about whether PHI is beneficial. The aim of this study was to estimate the effect of PHI in patients after sTBI. A multicenter, prospective cohort study was performed in Switzerland, including 832 adults with sTBI. Outcomes were death and impaired consciousness at 14 days. Associations between risk factors and outcomes were assessed with univariate and multivariate Cox models for survival, and univariate and multivariate regression models for impaired consciousness. Potential risk factors were age, GCS on scene, pupil reaction, Injury Severity Score (ISS), PHI, oxygen administration, and type of admission to trauma center. Age, GCS on scene < 9, abnormal pupil reaction and ISS ≥ 25 were associated with mortality. GCS < 9 and ISS ≥ 25 were correlated with impaired consciousness. PHI was overall not associated with short-term mortality and consciousness. However, there was a significative interaction with PHI and major trauma. PHI improves outcome from patients with sTBI and an ISS ≥ 25.

## 1. Introduction

Severe traumatic brain injury (sTBI) is a leading cause of death in developing and high-income countries [1,2,3,4] and represents a major socioeconomic burden due to permanent disability [3,5]. National reports have identified incidences between 4 and 20/100,000 persons [1,6,7], with a fatality rate of thirty percent [5,6]. Road traffic accidents and falls account for the majority of the cases [1,5,6,8] and given the reality of an aging population in Europe and the increase of the motorized travel in emerging countries, TBI will continue to be a significant health issue in the future [5].

Secondary systemic insults such as hypoxemia, hypotension, hyperthermia and hypo/hypercapnia exacerbate neuronal injury. The inefficient breathing of a comatose patient results in hypoxemia and hypercapnia. Hypoxemia decreases substrate delivery of oxygen to an injured brain, and hypercapnia leads to vasodilatation that could increase an intracranial hemorrhage. Those systemic insults are strongly associated with higher mortality rates. Pre-hospital airway management should focus on limiting those systemic insults, and PHI could precisely improve the management of hypoxemia and hypercapnia [9,10,11].

Hypoxemia is an independent factor shown to worsen the prognosis and should be treated aggressively even prior to hospital admission [6,12]. Adequate airway management and oxygenation in the pre-hospital setting is therefore of fundamental importance [10,13]. Commonly, a Glasgow Coma Scale (GCS) < 9 is identified as a threshold under which specific management is advised, such as intubation [14,15]. Pre-hospital intubation (PHI) offers the theoretical benefits of reducing hypoxia, controlling ventilation and end-tidal CO_2_, as well as protecting the airways [14,16]. However, reports suggest that PHI may participate in increasing the mortality rate for patients with TBI, due to a delay of hospital admission and/or a low success rate [17,18].

Pre-hospital assessment of the severity of injury of traumatic brain injury relies on various scores, the GCS [19] being the most commonly used to guide clinical management. The GCS has been proven to be a reliable tool to estimate prognosis [19], while its pessimistic overestimation of brain injury and high inter-individual variability confers limitations. The Head Abbreviated Injury Score (HAIS) provides useful prognosis information in patients with traumatic brain injury [20,21] while the Injury Severity Score (ISS) [22] provides an anatomical score for severity, but none of these latter aid the clinical management of the traumatic brain injury patient.

PHI for traumatic brain injury patients is subject to ongoing controversy. Most studies are observational and retrospective. Furthermore, out-of-hospital emergency systems vary in different countries and studies, with a likely effect on outcome. The aim of this study was to investigate the effect of pre-hospital intubation on outcome of patients having sustained a severe traumatic brain injury in Switzerland based on the severity of their injury.

## 2. Methods

### 2.1. Study Design

We conducted a nation-wide prospective epidemiological cohort study including the follow-up of patients with severe TBI from the time of event until their death, or until 14 days for survivors. This study is a part of the Swiss National Cohort of severe TBI entitled “Patient-relevant Endpoints after Brain Injury from Traumatic Accidents” (PEBITA; http://www.rettungsforum.ch/dynpg/upload/imgfile48.pdf). The detailed study protocol has been described elsewhere [6,12].

### 2.2. Participating Centers

The participating centers were 11 of the now 12 governmental defined trauma centers in Switzerland. The centers included Kantonsspital Aarau (KSA), Universitätsspital Basel (USB), Inselspital Bern, Kantonsspital Graubünden (KSGR), Hôpitaux Universitaires de Genève (HUG), Centre Hospitalier Universitaire Vaudois (CHUV), Kantonsspital Luzern (KSL), Hôpital de Sion (HVS), Kantonsspital St. Gallen (KSSG), Kantonsspital Winterthur (KSW), and Universitätsspital Zürich (USZ). Five centers are university hospitals.

The study took place from 1 May 2007 to 30 April 2010, with three centers participating throughout the entire period, three for a period of 6 months (from 1 November 2009 to 30 April 2010), and the other five for periods ranging from 6 months to 3 years (all ending on 30 April 2010).

### 2.3. Ethical Aspects

The study was approved by the ethics committees of the participating trauma centers (Ethics committee of Geneva, protocol NAC 07-013, approval: 7.5.2007). As a result of their neurological condition, patients were unable to give informed consent before enrolment. The local study coordinators contacted their legal representatives (proxies) to inform them of the study within 14 days following the TBI. Both patients and/or proxies received detailed written information on the study and were asked for consent. In the case of withdrawal, further follow-up was discontinued. Complying with a patient’s request, the collected data was removed from the database and destroyed.

### 2.4. Included Patients

All patients over 16 years old who sustained severe TBI from blunt or penetrating trauma and who were admitted to a Swiss trauma center were included. There was no upper limit for age. Severe TBI was defined by an Abbreviated Injury Scale score of the head region (HAIS) higher than 3, which was assessed clinically by neurosurgeons and based on computed tomography (CT) imaging findings. The worst CT scan in the first 24 h was assessed using a standardized data sheet based on the HAIS and Marshall classification [23,24]. Patients who died before neurosurgical or radiological diagnosis were included if the history of trauma and trauma signs were documented by out-of-hospital emergency medical systems (OHEMS). Patients were excluded if they had an unclear brain trauma history (for instance, comatose patients found in a public area without any bystanders), or if they had no signs of brain trauma (for instance, fatal multi-trauma patients with abdominal and thoracic injuries without visual injuries to the head). We did not use GCS as an inclusion criterion, due to its high inter-rater variability [25].

### 2.5. Data Collection

Trained collaborators from participating centers collected information using standardized data abstraction forms in paper format. Based on recommendation for TBI research, the patient data set was an adaptation of the Utstein-style documentation for traumatically injured patients in emergency medicine [26]. At all stages of data collection, study collaborators sought the advice of responsible physicians and/or qualified nurses for clarification in case of ambiguous or missing information. Prehospital care was investigated using standardized protocols including professional categories on scene and during transport. Specific data abstraction forms were used to guarantee high-quality data abstraction [26].

The following parameters were collected: patients’ demographic information, the GCS (3–15), the HAIS (1–6), the ISS (1–75) within the 24 h following injury, the presence of multiple trauma, and trauma mechanisms (falls, road traffic accidents, or others). Data also included the presence of hypotension (systolic blood pressure <90) and pupil reaction (normal/abnormal). Airway management data were collected, and included oxygen administration, intubation, end-tidal CO_2_ defined as normal between 28 and 45 mmHg, as well as the presence of hypoxemia defined as oxygen saturation below 90%. PHI was performed under the decision of a pre-hospital team doctor. The PHI protocol included GCS < 9 as criterion or an oxygen saturation below 90% despite oxygen therapy. PHI in Switzerland is always performed by a skilled doctor with a rapid sequence induction (usually succhinylcholine and etomidate or propofol, an opiate may be added to limit the rise in blood pressure). All data was collected specifically for the pre-hospital and in-hospital setting, when applicable. The type of admission to the trauma center (direct, indirect) and the pre-hospital time (in minutes) to hospital admission were also collected.

In order to assess the modification of PHI from a risk factor in a univariate model to a protective factor in a multivariate model, we included the interaction between severely injured patients and intubation (modification effect) in the univariate and multivariate analyses of surviving and non-surviving patients with sTBI at 14 days.

### 2.6. Outcome Measures

Mortality and GCS was collected at 14 days. GCS at 14 days has been shown to be a good surrogate for functional outcome, as a higher GCS at 15 days was associated with patients’ functional independence in survivors at 12 months [27].

### 2.7. Statistics

Qualitative variables such as sex, pupil reaction, HAIS, mechanism of accident, hypotension, hypoxemia, and type of admission to ED were summarized using percentages for the entire cohort. Quantitative variables such as age, GCS on scene, ISS, and time from OHEMS departure on scene to arrival in ED in the trauma center were described by their median and interquartile range (IQRs) (25th to 75th percentile) for the entire cohort.

Descriptive statistics were conducted for the following subgroups: survivors versus non-survivors at 14 days, and impaired consciousness (GCS ≤ 13) versus consciousness (GCS 14 and 15) for survivors at 14 days. Differences between two groups were assessed by non-parametric Wilcoxon rank-sum test with an alpha threshold of 5% for quantitative variables, and by Chi-square tests with an alpha threshold of 5% for qualitative variables.

In order to evaluate the association between pre-hospital risk factors and outcome at 14 days we performed a Cox regression (survival) or a logistic regression (impaired consciousness). Age, GCS on scene, pupil reaction, ISS, oxygen administration, direct or indirect admission to ED, and pre-hospital intubation were first included in a univariate Cox regression or logistic regression. Those data were then analyzed in a multivariate model. We analyzed a potential modification effect between PHI and severity of trauma, calculating the risk ratio for mortality due to PHI in patients severely injured or not (ISS < 25 or ≥ 25). For regression analyses, results were summarized as hazard ratios and 95% confidence intervals (CI) for the Cox model, and as odds ratios (OR) and 95% CI for the logistic model.

All statistical analyses were performed using STATA Release 14.2 (Stata Statistical Software: Release 14.2, Stata Corporation, College Station, TX, USA).

## 3. Results

The flowchart of included patients is shown in Figure 1.

Of the 832 patients, 73.6% of patients were male (Table 1). Isolated head trauma was present in 559 patients (67.2%). Mechanism of injury was falls in 432 patients (50.8%), and road traffic accidents in 277 cases (33.3%), while 15.9% (132) were due to other mechanisms, such as being wounded by objects, gunshots or sporting accidents. The majority of admissions were direct admissions (692 patients, 83.2%), with a mean pre-hospital time (scene to trauma center) to admission (direct or indirect) of 50 min (36–71).

Secondary insults were significantly more frequent in the non-survivors group (Table 2). While Oxygen was administered in more than 90% of the patients, significantly more hypoxemia was present in the non-survivor group, both in the pre and in-hospital setting. A similar difference was identified as significant, with more patients presenting a normal ETCO_2_ on scene or during transport in the survivors group (103, 80.5%) than in the non-survivors group (64, 55.2%). The ETCO_2_ in the emergency department was not statistically different between survivors and non-survivors groups. The number of missing values for ETCO_2_ was too high to be able to construe the ETCO_2_ as a risk factor in the univariate and multivariate analysis. PHI was performed more often in the non-surviving group (164, 64.3%) while hypoxemia and hypotension were significantly more present.

Data for surviving patients with regained consciousness (GCS 14 and 15) and impaired consciousness (GCS ≤ 13) are shown in Table 3. Isolated head injury was present in 252 (68.3%) of the cases. For overall surviving patients, falls were the mechanism of injury in 48% (177), and road traffic accidents in 35.5% (131), with other mechanisms were responsible for the remainders (61, 16.5%). Patients with falls as a mechanism of injury had a higher GCS at 14 days, while those having suffered from a road traffic accident were more prone to have an impaired consciousness at 14 days. The majority of admissions were direct (297 patients, 80.5%).

With a univariate regression model (Cox model), we found that age (1.02, 1.01–1.02), GCS < 9 (0.85, 0.83–0.88), abnormal pupil reaction on scene (3.9, 3.02–5.02), and ISS ≥ 25 were associated with mortality.

The percentage of episodes of hypotension and hypoxemia were globally small for surviving patients (Table 4). There were more episodes of hypoxemia in the patients with impaired consciousness at 14 days (in comparison with the group with regained consciousness). The percentage of PHI was also higher in this group.

Univariate and multivariate analyses of surviving and non-surviving patients with severe TBI at 14 days are shown in Table 5. There was an interaction between PHI and ISS. Compared to patients with an ISS < 25, risk of death was higher in patients with ISS ≥ 25 (HR = 7.59, *p* < 0.0001). Furthermore, PHI was also found to be a risk factor for death (HR = 2.83, *p* 0.066), but in the case of patients with an ISS > 25, this risk was mitigated by a four-fold interaction factor (HR = 0.25, *p* = 0.013) (Figure 2).

Univariate and multivariate analyses of surviving patients with regained consciousness (GCS 14 and 15) and impaired consciousness (GCS ≤ 13) at 14 days showed an association between a GCS < 9 and an ISS ≥ 25 with impaired consciousness at 14 days. All other parameters, including PHI, were not associated with this outcome.

## 4. Discussion

This study demonstrated that age, a GCS < 9, an abnormal pupil reaction and an ISS ≥ 25, were linked with mortality and impaired consciousness at 14 days after the head injury. However, the main finding of our study was that severe traumatic brain injury patients, defined as an ISS ≥ 25, benefitted from pre-hospital intubation, as it impacted positively on their survival. PHI was otherwise not statistically associated with outcome for any other patient-related variables.

The impact of PHI on outcome for patients with severe traumatic brain injury is controversial.

The 4th edition of the Brain Trauma Foundation’s guidelines suggest endotracheal intubation for any patient with a GCS ≤ 8, due to the compromised airway. Although a protected airway allows optimization of oxygenation and CO_2_ control, the procedure by itself carries inherent risks that might, in specific settings, outweigh its benefit. Furthermore, the GCS by itself has shown limitations in identifying the severity of the brain injury and may mislead the physician about the patient’s needs and priorities [28]. Nevertheless, hypoxia itself is not always the main reason for PHI [29], and this could be a reason for the negative outcomes associated with PHI found in some studies [30,31].

Pre-hospital intubation mandates a specific set of skills and training due to the environment and settings. While some studies found a favorable outcome of PHI for patients with severe brain traumatic injury, those referred to settings in which intubations were performed by highly-skilled healthcare providers [32]. In our setting, all pre-hospital intubations were performed by specifically trained physicians in anesthesia and emergency medicine. This could have reduced the risks of complications associated with PHI and therefore give an interesting insight into already performed studies in various settings; in particular in the United States or Australia, where PHI may be performed by paramedics without neuromuscular blockade or hypnotic drugs [33].

While PHI represents a specific set of skills, the procedure in Switzerland includes a pre-established pharmacological protocol [34]. Mandatory sedation, associated with pain control and paralysis, allows for proper management of the airway and control of the end-tidal CO_2_, therefore minimizing one of the main secondary insults. Unsedated patients with sTBI who are intubated show higher mortality in comparison to those who are not intubated, as shown by Lefering et al. in a study from 2017 [35]. PHI without sedation may lead to failed intubation and hypoxemia. Furthermore, intubation without sedation is often performed by paramedics who are usually less skilled in airway management than physicians.

While delays to hospital admission have been identified as a key player in relation to outcome, we were able to show no difference in pre-hospital time for survivors and non-survivors. Furthermore, due to the small size of our country, the time to hospital was relatively short in comparison to other systems.

Among the limitations of our study, we must acknowledge that it is a glimpse of a management of severe traumatic brain injured patients in a single country. Switzerland is characterized by various health systems, each under the authority of the respective canton. Disparities of treatment and management might be present, although a vast majority of patients included were finally managed by a nationally-operated, physician-staffed, pre-hospital system.

## 5. Conclusions

Pre-hospital intubation benefits patients who have sustained a severe traumatic brain injury as defined by a HAIS. PHI should be performed by a trained physician, associating regularly rehearsed skills to a standardized protocol. This will minimize the rate of complications and secondary insults prior to the arrival at a trauma center. Further studies should aim to define criteria and conditions (circumstances, situations) where PHI is detrimental and where it is necessary.

## Figures and Tables

**Figure 1 jcm-08-00470-f001:**
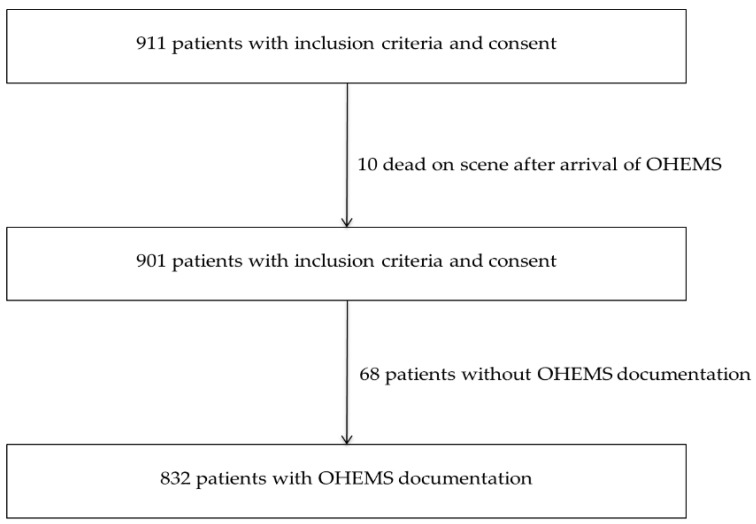
Flowchart of included patients with severe traumatic brain injury.

**Figure 2 jcm-08-00470-f002:**
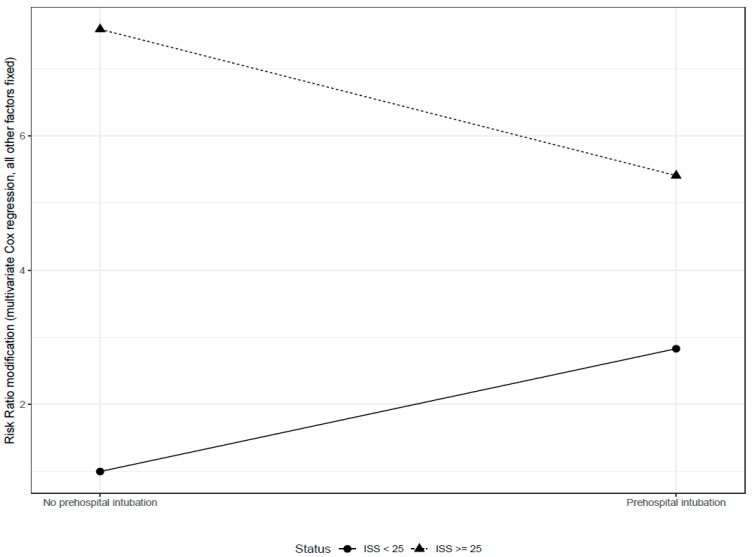
Interaction plot to illustrate the joint effect of pre-hospital intubation and Injury Severity Score on overall survival.

**Table 1 jcm-08-00470-t001:** Description of surviving and non-surviving patients with severe traumatic brain injury at 14 days.

Patient Characteristics	*n* Missing	*n* (%) *n* = 832	Survivors *n* = 577	Non-Survivors *n* = 255	*p* Value (Univariate Analysis)
Female	0	220 (26.4%)	150 (26.0%)	70 (27.5%)	0.661 ^A^
Age (years; median, IQR)	0	54.3 (32.2–71.3)	49.8 (28.4–67.5)	63.3 (41.6–79.5)	<0.0001 ^Δ^
GCS on scene (median, IQR)	11	9 (4–14)	12 (7–14)	4 (3–9)	<0.0001 ^Δ^
Abnormal pupil reaction	63	196 (25.5%)	69 (13.0%)	127 (52.9%)	<0.0001 ^A^
HAIS 4	0	357 (42.9%)	320 (55.5%)	37 (14.5%)	<0.0001 ^A^
HAIS 5	0	448 (53.9%)	257 (44.5%)	191 (74.9%)	
HAIS 6	0	27 (3.3%)	0	27 (10.6%)	
ISS (median, IQR)	0	25 (21–34)	25 (20–30)	29 (25–41)	<0.0001 ^Δ^
Pre-hospital time	137	50 (36–71)	50 (37–75)	48 (35–65)	0.0977 ^Δ^

^A^ Chi-square test. ^Δ^ Wilcoxon rank-sum-test.

**Table 2 jcm-08-00470-t002:** Secondary insults and effect on outcome at 14 days.

Parameter	*n* Missing	*n* (%) *n* = 832	Survivors *n* = 577	Non-Survivors *n* = 255	Odds Ratio ^Γ^ (95% CI)	*p* Value (Univariate Analysis)
Hypoxemia sc/tr	192	100 (15.6%)	53 (11.7%)	47 (25.1%)	2.53 (1.64–3.92)	<0.0001
PHI	0	367 (44.1%)	203 (35.2%)	164 (64.3%)	3.32 (2.44–4.52)	<0.0001
Without PHI	0	465 (55.9%)	374 (64.8%)	91 (35.7%)	0.30 (0.22–0.41)	<0.0001
Hypoxemia ED	41	45 (5.7%)	12 (2.2%)	33 (13.8%)	7.12 (3.61–14.05)	<0.0001
Hypotension ED	20	39 (4.3%)	8 (1.4%)	31 (12.7%)	10.24 (4.64–22.64)	<0.0001

Hypoxemia = SpO2 < 90, sc/tr = on scene or during transport, PHI = pre-hospital intubation, hypotension = systolic blood pressure < 90, ED = emergency department. ^Γ^ for outcome «non survivors», *p* value against OR = 1.

**Table 3 jcm-08-00470-t003:** Description of surviving patients with regained consciousness (Glasgow Coma Scale 14 and 15) and impaired consciousness (Glasgow Coma Scale ≤ 13) at 14 days.

Patient Characteristics	*n* Missing	*n* (%) *n* = 369	GSC 14/15 *n* = 267	GCS ≤ 13 *n* = 102	*p* Value (Univariate Analysis)
Female	0	95 (25.8%)	74 (27.7%)	21 (20.6%)	0.161^A^
Age (years; median, IQR)	0	50.2 (29.9–66.7)	52.1 (32.8–69.3)	45.8 (24.6–63.2)	0.0261^Δ^
GCS on scene (median, IQR)	0	11 (6–14)	13 (9–14)	6 (4–10)	<0.0001^Δ^
Abnormal pupil reaction	36	46 (13.8%)	21 (8.5%)	25 (28.7%)	<0.0001^A^
ISS (median, IQR)	0	25 (20–33)	25 (17–29)	29 (25–38)	<0.0001^Δ^
HAIS 4	0	197 (53.4%)	166 (62.2%)	31 (30.4%)	<0.0001^A^
HAIS 5	0	172	101	71	<0.0001^Δ^

^A^ Chi-square test. ^Δ^ Wilcoxon rank-sum-test.

**Table 4 jcm-08-00470-t004:** Parameters of surviving patients with regained consciousness (Glasgow Coma Scale 14 and 15) and impaired consciousness (Glasgow Coma Scale ≤ 13) at 14 days.

Parameter	*n* Missing	*n* (%) *n* =369	GSC 14/15 *n* = 267	GCS ≤ 13 *n* = 102	Odds Ratio ^Γ^ (95% CI)	*p* Value (Univariate Analysis)
Hypotension sc/tr	70	12 (4.0%)	7 (3.4%)	5 (5.6%)	1.68 (0.52–5.46)	0.373
Hypoxemia sc/tr	74	34 (11.9%)	12 (5.9%)	22 (26.8%)	5.84 (2.73–12.49)	<0.0001
PHI	0	139 (37.7%)	66 (24.7%)	73 (71.6%)	7.67 (4.59–12.80)	<0.0001
Without PHI	0	230 (62.3%)	201 (75.3%)	29 (28.4)	0.13 (0.08–0.22)	<0.0001
Hypoxemia in ED	6	9	6 (2.3%)	3 (2.9%)	1.27 (0.31–5.18)	0.732
Hypotension in ED	0	5 (1.4%)	4 (1.5%)	1 (0.98%)	0.65 (0.07–5.89)	0.698

Hypoxemia = SpO2 < 90, sc/tr = on scene or during transport, PHI = pre-hospital intubation, hypotension = systolic blood pressure <90, ED = emergency department. ^Γ^ for outcome «non survivors», *p* value against OR = 1.

**Table 5 jcm-08-00470-t005:** Univariate and multivariate analyses of surviving and non-surviving patients with severe traumatic brain injury at 14 days including the interaction between severely injured patients and intubation (modification effect).

Initial Trauma Assessment	Univariate Regression Model (Cox model); Hazard Ratio (95% CI)	*p* Value	Multivariate Regression Model (Cox Model with BRESLOW Method for Ties); Hazard Ratio ^Ω^ (95% CI)	*p* Value
Age (years)	1.02 (1.01–1.02)	<0.0001	1.02 (1.02–1.03)	<0.0001
GCS < 9 on scene	3.29 (2.48–4.35)	<0.0001	2.44 (1.60–3.72)	<0.0001
Abnormal pupil reaction on scene	3.90 (3.02–5.02)	<0.0001	2.54 (1.87–3.47)	<0.0001
ISS ≥ 25	4.67 (2.96–7.37)	<0.0001	7.59 (3.27–17.62)	<0.0001
**Pre-hospital procedures**				
Intubation	2.27 (1.75–2.93)	<0.0001	2.83 (0.93–8.56)	0.066
Interaction: ISS ≥ 25 * Intubation			0.25 (0.08–0.74)	0.013
Oxygen administration	0.46 (0.23–0.89)	0.021	0.87 (0.36–2.07)	0.751

^Ω^ For outcome “non-survivors”, *p* value against HR = 1.

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
