# Peer review of "Effect of Pre-Hospital Intubation in Patients with Severe Traumatic Brain Injury on Outcome: A Prospective Cohort Study"

_jcm, 2019, doi:10.3390/jcm8040470_

Reviewer 1 Report

Effect of prehospital intubation in patients with severe traumatic brain injury on outcome:  a prospective cohort study.

Jcm-471315

The study is a prospective study looking at the impact of prehospital intubation in patients after sTBI. Various risk factors an outcomes were utilized in this study.

Abstract

1.       Period is off on the fourth to last sentence

2.       The third to last sentence is confusing and needs to re-worded to not end with “…PHI, not.

Intro

1.       Line 46, “it has been proven to be a reliable..” should say “The GCS has been shown to be reliable tool..”

2.       Line 52, the aim is listed as a separate paragraph. I recommend adding a few summarizing sentences of the current research/limitation and then listing the aim as the concluding sentence of that paragraph.

Methods

1.       Lines 56-57, the first sentence is confusing. Please clarify “…from the time of event until 14 days or earlier death.”

2.       I recommend giving more details on the measures used including number of items, ranges and severity of scores, and any psychometric data.

3.       Although age is included in the demographic table, it would be helpful Include age ranges in the methods section, line 82.

Results

1.       Table 1 is a bit hard to follow. I recommend one row per characteristic. The values in the second row 54.3 and 49.8 are hard to follow and I’m not sure what they represent. I would include means and SD for age, GCS, and ISS.

2.       Table 2 and 3 the first column is hard to follow with the extra short lines.

Discussion

1.       Line 196 seems out of place and does not flow. There needs to be better transition throughout the discussion.

Conclusion

2.       Line 228 “It should be…” Avoid using it at the start of a sentence because it is not always clear what “it” is representing. Instead state “PHI should be performed….”

Overall, the study is a useful contribution to the literature, but needs some editing including flow across paragraphs and some minor changes in sentence structure.

Author Response

Thank you very much for your comments.

Abstract. 

I don't quite understand what you mean with "period is off on the fourth to last sentence"?

I agree that is maybe not appropriate, I rewrote and shortened the abstract a little bit and I hope it's better.

Introduction.

I corrected it.

You are right, I pointed out some of the limitations of the studies already published. 

Methods. 

I corrected it to make it more clear and hope it is.

You are also right it is better with ranges so that the reader understand it correctly. I modified it according to your comment. 

There was no upper limit for age. I added this specification in the text.

Results. 

I totally agree, so I've redone the tables, and I hope that suits you.

Regarding the GCS and ISS scores, we decided to present the median and IQR as the values were not distributed normally. Therefore, we feel that the median better represents the population in that particular case. Regarding Age, the mean would probably be appropriate, but for consistency reasons the median is also presented. 

Discussion. 

I modified the beginning of the discussion according to your comment and thank you for that. I hope the transition is better now. 

I corrected it.

Thank you again for your interesting comments, so that I could improve our article. 

Regards,

Caroline Choffat

Reviewer 2 Report

This is an interesting article, that addresses an important issue for emergency medicine. However there are quite a few limitations in the manuscript that need to be addressed.

Abstract

Some language and grammar improvements are needed in: line 9, 20 - 21 of the abstract. The Also the acronym for GCS is not presented.

A key finding of the study is not clear in the abstract, lines 20 - 21.

Introduction

lines 28 to 29 are uniformative without more specifics about demographics, such as percentages.

Lines 33 to 34 report that the secondary processes are established and important, but do not state why. Some mechanisms and insights are essential here to understand why PHI may be considered.

Line 38 to 40 is unclear. Please correct grammar or clarify what is meant.

Line 47: the Crash Trial appears to have inconsistent referencing compared to the other references (i.e. not numerical).

Methods

Considering the importance of PHI to this study, the criteria and use of PHI is not reported adequately. In particular what is considered prehospital/inhospital, how it is administered. And perhaps most importantly, what criteria are used for intubation. The process of intubation is most essential to interpreting findings.

The differences between trauma centres is also not explored.

Results

The figure 1 needs to be substantially improved, including the removal of formatting mark-up, full wording for the abbreviations in the figure legend.

The statistical results are insufficient. Many tables are unclear what the p-value means, and there is no measure of effect size. These should be reported in the tables, even if some are reported in text. The table legends also need to be expanded so the they can be interpreted better. In addition for the most important outcomes, especially PHI, there should be graphs of the data so the readers can interpret the findings better.

Some discussion about the differences between hospitals would be useful.

Discussion

The study has an important outcome. The discussion regarding different levels of competence and resource settings is useful. The protocol for PHI in Switzerland and how it it may differ from other countries would be worth discussing further.

Author Response

Thank you very much for your comments. I hope the modifications will suit you, and I am open to any other comments.

Abstract.

I added the acronym for the Glasgow Coma Scale.

I modified the abstract to make it more concise and clear, according to your comment. 

Introduction. 

I agree with you that percentages are important so I added that. 

You are right, some insights about physiopathology can help the reader understand why treating secondary insults is so important, so I modified my text according to that point. I hope it is more clear and precise for the definitions. 

    3. I corrected it thanks to your comment. 

    4. I found another reference that is more appropriate. 

Methods.

Your comment is relevant. I included some insights on the use of GCS as a criterion for intubation and on its limitations. I added some details about protocols for prehospital intubation in Switzerland. I hope it is now more clear for the reader and thank you for your comment. 

Results.

I modified the flowchart according to your comment.

Thank you for that comment. My goal is to make it clear, so I modified the tables according to your comment. I added precisions about the p-values in the table legends. We also calculate odds ratios for table 2. and table 4. to permit to the reader a measure of effect size. In addition and I thank you for that, we added a graph so the reader can understand better the importance of the interaction of prehospital intubation and ISS. I also gave some insights about that point in the text. Thank you for your comment. 

Yes I agree, I specified that 5 hospitals from the participating centers are universitary hospitals, which could be said as equivalent to trauma centers, other participating centers being peripheral hospitals with less skills to take charge of trauma patients. 

Discussion.

It is indeed important and I added in the discussion some differences between Switzerland and other countries where prehospital intubation is not always performed by physicians. I also gave precisions about protocols of intubation in Switzerland. We do not have one global protocol for the whole country, as intubation is always performed by a skilled physician who will decide whether he uses one drug or the other. Nevertheless, intubation will always be performed with an hypnotic drug and neuromuscular blockade.

Thank you again for all your comments, 

Regards,

Caroline Choffat

Round  2

Reviewer 2 Report

I am happy with the changes and appreciate the responses.